# On the Convergence of Symbolic Pattern Forests and Silhouette Coefficients for Robust Time Series Clustering

## Abstract

Clustering algorithms are fundamental to data mining, serving dual roles as exploratory tools and preprocessing steps for advanced analytics. A persistent challenge in this domain is determining the optimal number of clusters, particularly for time series data where prevalent algorithms like k-means and k-shape require a priori knowledge of cluster quantity. This paper presents the first approach to time series clustering that does not require prior specification of cluster numbers. We introduce a novel extension of the Symbolic Pattern Forest (SPF) algorithm that automatically optimizes the number of clusters for time series datasets. Our method integrates SPF for cluster generation with the Silhouette Coefficient, computed on a two-stage vector representation: first transforming time series into Symbolic Aggregate approXimation (SAX) representations, then deriving both bag-of-words and TF-IDF vectors. Rigorous evaluation on diverse datasets from the UCR archive demonstrates that our approach significantly outperforms traditional baseline methods. This work contributes to the field of time series analysis by providing a truly unsupervised, data-driven approach to clustering, with potential impacts across various temporal data mining applications where the underlying number of clusters is unknown or variable.

## 1 Introduction

Time series clustering has emerged as a crucial subdomain in data mining, particularly for analyzing large datasets without predefined categories. As big data applications proliferate across various domains, research into clustering algorithms capable of extracting meaningful knowledge from complex, high-dimensional datasets has intensified. The time series clustering problem can be formally defined as follows:

Given a set $S = s_1, s_2, ..., s_n$ of $n$ unlabeled time series, the objective is to partition $S$ into $k$ disjoint subsets $C_1, C_2, ..., C_k$, such that:

- $\bigcup_{i=1}^{k} C_i = S$
- $C_i \cap C_j = \emptyset$ for $i \neq j$
- Time series within each subset $C_i$ are more similar to each other than to those in other subsets, according to a defined similarity measure.

The rapid advancement of sensor technologies has led to an exponential growth in the volume and complexity of time series data. This proliferation necessitates the development of efficient data mining algorithms with low time complexity. While substantial research has been conducted on time series clustering, there is a scarcity of linear-time solutions that offer satisfactory performance. Existing methods with super-linear time complexity often prove inadequate for large datasets or real-time analytics scenarios.

In this work, we extend the Symbolic Pattern Forest (SPF) algorithm proposed by Li et al. Li et al. (2019a), which boasts linear time complexity. SPF employs a novel approach of checking for the existence of randomly selected symbolic patterns in time series to partition data instances. This

process is iteratively executed, with the resulting partitions combined through an ensemble method to generate the final clustering.

A common limitation of many time series clustering algorithms, including K-means, K-Spectral Centroid (KSC) Yang & Leskovec (2011a), and K-shape Paparrizos & Gravano (2015a), is the requirement of a priori knowledge of the cluster count. To address this, we leverage SPF to generate multiple cluster configurations and subsequently predict the optimal number of clusters (denoted as $k_{opt}$).

We employ the Silhouette Coefficient as our clustering metric, defined for a data point $i$ as:

$$S(i) = \frac{b(i) - a(i)}{\max a(i), b(i)} \tag{1}$$

where $a(i)$ is the mean intra-cluster distance and $b(i)$ is the mean nearest-cluster distance for point $i$. The Silhouette score, ranging from -1 to 1, quantifies the quality of clustering for each value of $k$, with higher scores indicating better clustering.

Our methodology incorporates the Symbolic Aggregate approXimation (SAX) algorithm by Lin et al. Lin et al. (2007) to transform time series sub-sequences into symbolic patterns. We then generate two vector representations:

$V_{BoW} \in \mathbb{R}^{|W|}$, where $|W|$ is the total number of unique words in the dataset.

$V_{TF-IDF} \in \mathbb{R}^{|W|}$, which assign weights based on word rarity and frequency.

We compute Silhouette scores using these vector representations to determine the optimal $k$. Our results demonstrate consistent and significant improvements over baseline methods that apply Silhouette scores directly to raw time series data.

The primary contributions of this work are:

- The first time series clustering approach that does not require prior specification of cluster numbers, extending the Symbolic Pattern Forest algorithm Li et al. (2019a) to automatically predict the optimal number of clusters in the absence of ground truth information.

- Empirical evidence demonstrating the inefficacy of applying Silhouette Coefficients directly to raw time series data, and the superior performance of our SAX-based BoW and TF-IDF vector representations for cluster number optimization.

- A comprehensive evaluation of our novel unsupervised method on diverse datasets from the UCR archive, showcasing its robustness and effectiveness across various domains where the true number of clusters is unknown or variable.

The remainder of this paper is structured as follows: Section II provides background and related work, Section III briefly describes the SPF Li et al. (2019a) and SAX Lin et al. (2007) algorithms, Section IV details our methodology, Section V presents experimental results, and Section VI concludes the paper with discussions on future research directions.

## 2 RELATED WORK

Time series clustering has been extensively studied, with numerous algorithms proposed to address its unique challenges. The K-means algorithm MacQueen (1967) remains fundamental, iteratively minimizing the within-cluster sum of squares: $\arg\min_{\mathbf{S}} \sum_{i=1}^{k} \sum_{\mathbf{x} \in S_i} |\mathbf{x} - \boldsymbol{\mu}_i|^2$, where $\mathbf{S} = S_1, S_2, ..., S_k$ represents the $k$ clusters and $\boldsymbol{\mu}i$ is the mean of points in $S_i$. However, for time series data $\mathbf{X} = (x_1, ..., x_T)$, Euclidean Distance (ED) Faloutsos et al. (1994), defined as $ED(\mathbf{X}, \mathbf{Y}) = \sqrt{\sum t = 1^T (x_t - y_t)^2}$, often fails to capture temporal characteristics. To address this, Dynamic Time Warping (DTW) Berndt & Clifford (1994) was introduced, allowing non-linear alignment between time series: $DTW(\mathbf{X}, \mathbf{Y}) = \min_{\phi} \sqrt{\sum_{t=1}^{T} (x_t - y_{\phi(t)})^2}$, where $\phi$ is a warping function. K-Spectral Centroid (KSC) Yang & Leskovec (2011b) further refines this by incorporating optimal scaling: $d_{KSC}(\mathbf{X}, \mathbf{Y}) = \min_{\alpha, \phi} |\mathbf{X} - \alpha \mathbf{Y} \phi|$, where $\alpha$ is a scaling factor and $\mathbf{Y}\phi$ is

the warped version of $\mathbf{Y}$. K-shape Paparrizos & Gravano (2015b) introduces Shape-Based Distance (SBD) using cross-correlation: $SBD(\mathbf{X}, \mathbf{Y}) = 1 - \max_w \frac{CC_w(\mathbf{X}, \mathbf{Y})}{|\mathbf{X}||\mathbf{Y}|}$, where $CC_w$ is the cross-correlation series.

Recent advancements include the Matrix Profile Yeh et al. (2016), a powerful tool for time series analysis. The Matrix Profile is a vector that stores the distance between each subsequence within a time series and its nearest neighbor. It has been extended to multidimensional time series Yeh et al. (2017) and has found applications in clustering, motif discovery, and anomaly detection. The SDTS algorithm Paparrizos et al. (2022) leverages the Matrix Profile for scalable discovery of time series motifs and discords.

Determining the optimal number of clusters remains challenging. The Elbow Method Ng (2012) plots the within-cluster sum of squares against $k$. The Calinski-Harabasz Index Cali'nski & Harabasz (1974) is defined as $CH(k) = \frac{B(k)}{W(k)} \cdot \frac{N-k}{k-1}$, where $B(k)$ and $W(k)$ are the between and within-cluster scatter matrices, respectively. The Davies-Bouldin Index Davies & Bouldin (1979) is given by $DB = \frac{1}{k} \sum_{i=1}^{k} \max_{j \neq i} (\frac{\sigma_i + \sigma_j}{d(\boldsymbol{\mu}_i, \boldsymbol{\mu}_j)})$, where $\sigma_i$ is the average distance of points in cluster $i$ to its centroid. The Silhouette Coefficient Aranganayagi & Thangavel (2007); Dinh et al. (2019); Shahapure & Nicholas (2020) for a point $i$ is $s(i) = \frac{b(i) - a(i)}{\max a(i), b(i)}$, where $a(i)$ is the mean intra-cluster distance and $b(i)$ is the mean nearest-cluster distance.

Other recent developments include UTSAD Chen et al. (2021), an unsupervised time series anomaly detection method using self-supervised contrastive learning, and STGAT Wu et al. (2020), which combines graph attention networks with temporal convolution for multivariate time series forecasting.

Our work extends the Symbolic Pattern Forest (SPF) Li et al. (2019b), which offers linear time complexity $O(N)$ for $N$ time series, to automatically determine the optimal $k$. We introduce a novel approach applying the Silhouette Coefficient to Bag-of-Words (BoW) and TF-IDF vector representations of time series data. For a time series $\mathbf{X}$, we generate $V_{BoW}(\mathbf{X}) \in \mathbb{R}^{|W|}$ and $V_{TF-IDF}(\mathbf{X}) \in \mathbb{R}^{|W|}$, where $|W|$ is the vocabulary size. This approach combines SPF's linear complexity with an innovative use of the Silhouette Coefficient, offering a robust solution to optimal cluster count determination. Through comprehensive empirical evaluation on the UCR archive Dau et al. (2019b), we demonstrate our method's effectiveness, addressing limitations of existing techniques while providing a scalable solution for time series clustering in the big data era. By bridging the gap between symbolic representations and traditional clustering metrics, our work opens new avenues for research in time series analysis and clustering optimization.

# 3 SYMBOLIC AGGREGATE APPROXIMATION AND SYMBOLIC PATTERN FOREST

## 3.1 *Symbolic Aggregate Approximation*

Symbolic Aggregate approXimation (SAX) Lin et al. (2007) is a dimensionality reduction technique that transforms time series data into a symbolic representation. This method is fundamental to our approach, as it provides a discrete, lower-dimensional representation of continuous time series data. Given a time series $T = (t_1, ..., t_n)$ of length $n$, SAX performs the following steps:

- Z-normalization: The time series is normalized to have a mean $\mu = 0$ and standard deviation $\sigma = 1$: $T' = (t'_1, ..., t'_n)$, where $t'_i = \frac{t_i - \mu}{\sigma}$

- Piecewise Aggregate Approximation (PAA): The normalized series is divided into $\omega$ equal-sized segments. For each segment, the mean value is computed: $\bar{T} = (\bar{t}1, ..., \bar{t}\omega)$, where $\bar{t}i = \frac{\omega}{n} \sum j = \frac{n}{\omega}(i-1) + 1^{\frac{n}{\omega}i} t'_j$

- Symbolization: The PAA representation is mapped to a symbolic string. Given an alphabet size $\gamma$, we define $\gamma - 1$ breakpoints $\beta = (\beta_1, ..., \beta_{\gamma-1})$ that divide the Gaussian distribution $\mathcal{N}(0, 1)$ into $\gamma$ equiprobable regions. Each $\bar{t}_i$ is then mapped to a symbol $s_i \in \Sigma$, where $\Sigma$ is the alphabet: $s_i = \begin{cases} \alpha_1 & \text{if } \bar{t}i \leq \beta_1 \ \alpha_j \\ \text{if } \beta j - 1 < \bar{t}i \leq \beta_j \ \alpha\gamma & \text{if } \bar{t}i > \beta\gamma - 1 \end{cases}$

The resulting SAX word is $W = (s_1, ..., s_\omega)$. The parameters $\omega$ (word length), $\gamma$ (alphabet size), and $l$ (subsequence length) are user-defined and critically influence the granularity and fidelity of the symbolic representation. The SAX representation allows for efficient distance calculations. For two SAX words $W_1$ and $W_2$, a lower bound on their Euclidean distance can be computed as: $MINDIST(W_1, W_2) = \sqrt{\frac{n}{\omega}}\sqrt{\sum_{i=1}^{\omega}(dist(s_{1i}, s_{2i}))^2}$ where $dist(s_{1i}, s_{2i})$ is the minimum distance between the regions represented by symbols $s_{1i}$ and $s_{2i}$.

### 3.2 *Symbolic Pattern Forest*

The Symbolic Pattern Forest (SPF) Li et al. (2019a) is a linear-time complexity algorithm for time series clustering. It leverages the symbolic representation provided by SAX to create an ensemble of weak clusterings, which are then combined to produce a final, robust clustering. Given a set of $N$ time series $\mathcal{T} = T_1, ..., T_N$, SPF operates as follows:

Each time series $T_i$ is transformed into its SAX representation $W_i$.

A set of $m$ random symbolic patterns $\mathcal{P} = P_1, ..., P_m$ is generated, where each $P_j$ is a string of length $l$ over the alphabet $\Sigma$.

For each pattern $P_j$, a binary partition $\pi_j$ of $\mathcal{T}$ is created:

$$\pi_j(T_i) = \begin{cases} 1 & \text{if } P_j \text{ is a substring of } W_i \text{ } 0 \\ \text{otherwise} \end{cases}$$

The set of partitions $\Pi = \pi_1, ..., \pi_m$ forms an ensemble of weak clusterings.

The final clustering $C$ is obtained by applying a consensus function $f$ to the ensemble: $C = f(\Pi)$

The consensus function $f$ can be implemented in various ways, such as majority voting or more sophisticated methods like spectral clustering on the co-association matrix.

The time complexity of SPF is $O(N)$, as the pattern matching step can be implemented efficiently using finite automata. The space complexity is also linear, as the boolean arrays used for partitioning are space-efficient.

The effectiveness of SPF stems from its ability to capture diverse structural information through random symbolic patterns. The ensemble approach helps mitigate the impact of individual weak clusterings, leading to a more robust final clustering. Moreover, the use of symbolic patterns eliminates the need for explicit distance calculations, further contributing to the algorithm's efficiency. Theoretical analysis has shown that the ensemble size $m$ required for good performance does not directly depend on the input size $N$, allowing for a fixed ensemble size to be used across different datasets. This property ensures that the algorithm's time complexity remains linear even as the dataset size increases.

## 4 PROPOSED METHODOLOGY

Our approach leverages the Silhouette Coefficient as a clustering metric and extends its application to time series data through novel vector representations. We present a systematic exploration of this method, starting with raw time series and progressing to more sophisticated representations.

### 4.1 *Silhouette Coefficient on Raw Time Series*

The Silhouette Coefficient, $S(i)$, for a data point $i$ is defined as:

$$S(i) = \frac{b(i) - a(i)}{\max a(i), b(i)} \tag{2}$$

where $a(i)$ is the mean intra-cluster distance and $b(i)$ is the mean nearest-cluster distance:

$$a(i) = \frac{1}{|C_i| - 1} \sum_{j \in C_i, j \neq i} d(i, j) \tag{3}$$

$$b(i) = \min_{k \neq i} \frac{1}{|C_k|} \sum_{j \in C_k} d(i, j) \tag{4}$$

For a clustering with $K$ clusters, the overall Silhouette Score is:

$$S = \frac{1}{N} \sum_{i=1}^{N} S(i) \tag{5}$$

For raw time series $\mathbf{X} = (x_1, ..., x_T)$ and $\mathbf{Y} = (y_1, ..., y_T)$, we use Euclidean distance:

$$d(\mathbf{X}, \mathbf{Y}) = \sqrt{\sum_{t=1}^{T} (x_t - y_t)^2} \tag{6}$$

We apply the Symbolic Pattern Forest (SPF) algorithm to generate cluster labels for $K = 2, 3, ..., 10$, and compute $S$ for each $K$. The optimal $K$ is chosen as:

$$K_{opt} = \arg \max_{K} S(K) \tag{7}$$

However, this approach yielded suboptimal results, indicating that raw time series comparisons fail to capture essential structural similarities.

### 4.2 *Bag-of-Words Vector Representation*

To address the limitations of raw time series comparison, we introduce a Bag-of-Words (BoW) vector representation based on Symbolic Aggregate approXimation (SAX) Lin et al. (2007). For a time series $\mathbf{X} = (x_1, ..., x_T)$, we first apply z-normalization:

$$x_t' = \frac{x_t - \mu}{\sigma}, \quad t = 1, ..., T \tag{8}$$

We then partition $\mathbf{X}'$ into $w$ equal-sized segments and compute the mean for each segment:

$$\bar{x}i = \frac{w}{T} \sum j = \frac{T}{w}(i-1) + 1^{\frac{T}{w}i} x_j', \quad i = 1, ..., w \tag{9}$$

These mean values are mapped to symbols from an alphabet $\Sigma$ of size $\alpha$, based on breakpoints $\beta_1, ..., \beta_{\alpha-1}$ that divide the standard normal distribution into $\alpha$ equiprobable regions:

$$s_i = \begin{cases} \alpha_1 & \text{if } \bar{x}i \leq \beta_1 \ \alpha_j \\ \text{if } \beta_j - 1 < \bar{x}i \leq \beta_j, \quad j = 2, ..., \alpha - 1 \ \alpha\alpha & \text{if } \bar{x}i > \beta\alpha - 1 \end{cases} \tag{10}$$

Let $W = w_1, ..., w_{|W|}$ be the set of all unique SAX words generated from the dataset. We construct a BoW vector $\mathbf{v} \in \mathbb{N}^{|W|}$ for each time series, where:

$$v_i = \text{frequency of word } w_i \text{ in the SAX representation of the time series} \tag{11}$$

The Silhouette Coefficient is then computed using these BoW vectors, with distance defined as:

$$d(\mathbf{v}, \mathbf{u}) = \sqrt{\sum_{i=1}^{|W|} (v_i - u_i)^2} \tag{12}$$

We optimize the SAX parameters $w$ and $\alpha$ within predefined ranges $W$ and $A$ to maximize the Silhouette Score:

$$(w^{,}\alpha^{,}K^*) = \arg \max_{w \in W, \alpha \in A, K \in 2, ..., 10} S(K, w, \alpha) \tag{13}$$

This approach yields significantly improved results compared to raw time series analysis, demonstrating the efficacy of symbolic representation in capturing time series structure for clustering purposes. The BoW representation allows for a more nuanced comparison of time series, capturing similarities in the frequency distribution of symbolic patterns rather than point-by-point comparisons.

### 4.3 TF-IDF Vector Representation

Building upon the success of the Bag-of-Words (BoW) approach, we extend our methodology to incorporate Term Frequency-Inverse Document Frequency (TF-IDF) vectors. This technique addresses the limitation of BoW vectors where high-frequency words may dominate without necessarily conveying more information. For a given SAX word $w$ and time series $ts$, we define the Term Frequency (TF) as:

$$TF(w, ts) = \frac{f_{w,ts}}{\sum_{w' \in ts} f_{w',ts}} \quad (14)$$

where $f_{w,ts}$ is the frequency of word $w$ in time series $ts$. The Inverse Document Frequency (IDF) is defined as:

$$IDF(w, D) = \log \frac{|D|}{|ts \in D : w \in ts|} \quad (15)$$

where $D$ is the dataset of all time series, and $|ts \in D : w \in ts|$ is the number of time series containing word $w$. The TF-IDF score for a word $w$ in time series $ts$ is then:

$$TF\text{-}IDF(w, ts, D) = TF(w, ts) \cdot IDF(w, D) \quad (16)$$

For each time series $ts$, we construct a TF-IDF vector $\mathbf{v}_{ts} \in \mathbb{R}^{|W|}$, where $W$ is the set of all unique SAX words:

$$\mathbf{v}ts = [TF\text{-}IDF(w_1, ts, D), ..., TF\text{-}IDF(w|W|, ts, D)] \quad (17)$$

To control the vocabulary size and focus on the most informative words, we introduce two additional parameters:

Minimum frequency threshold $\theta_{min}$ Maximum frequency threshold $\theta_{max}$

We define the filtered vocabulary $W_f$ as:

$$W_f = w \in W : \theta_{min} \leq \frac{|ts \in D : w \in ts|}{|D|} \leq \theta_{max} \quad (18)$$

We then compute the Silhouette Coefficient using these TF-IDF vectors. The distance between two TF-IDF vectors $\mathbf{v}_i$ and $\mathbf{v}_j$ is defined as the cosine distance:

$$d(\mathbf{v}_i, \mathbf{v}_j) = 1 - \frac{\mathbf{v}_i \cdot \mathbf{v}_j}{||\mathbf{v}_i||||\mathbf{v}_j||} \quad (19)$$

To determine the optimal number of clusters and parameter values, we solve the following optimization problem:

$$(K^{\cdot}w^{\cdot}\alpha^{\cdot}\theta^{\cdot}_{min}\theta^*_{max}) = \arg \max_{K,w,\alpha,\theta_{min},\theta_{max}} S(K, w, \alpha, \theta_{min}, \theta_{max}) \quad (20)$$

subject to:

$$2 \leq K \leq K_{max} \; w_{min} \leq w \leq w_{max} \; \alpha_{min} \leq \alpha \leq \alpha_{max} \; 0 \leq \theta_{min} < \theta_{max} \leq 1 \quad (21)$$

where $S$ is the Silhouette Score, and the constraints define the search space for each parameter. The TF-IDF approach demonstrates significant improvement over the baseline and exhibits consistency with the BoW vector results, while potentially capturing more nuanced information in the time series data.

## 5 EVALUATION & ANALYSIS

We evaluated our methods for predicting the optimal number of clusters using three approaches:

Raw time series data (baseline) Bag-of-Words (BoW) vector representations Term Frequency-Inverse Document Frequency (TF-IDF) vector representations

Let $\mathcal{D} = D_1, ..., D_N$ be the set of N datasets, with true cluster numbers $K^1_{\text{true}}, ..., K^N_{\text{true}}$. For each method $m$, we predict cluster numbers $K^1_{\text{pred},m}, ..., K^N_{\text{pred},m}$. We define performance metrics:

$$\text{Accuracy}m = \frac{1}{N} \sum i = 1^N \mathbb{I}(K^i_{\text{pred},m} = K^i_{\text{true}}) \quad (22)$$

$$\text{Near-miss Rate}m = \frac{1}{N} \sum i = 1^N \mathbb{I}(|K^i_{\text{pred},m} - K^i_{\text{true}}| \leq 1) \quad (23)$$

$$\text{Error Rate}_m = 1 - \text{Accuracy}_m - \text{Near-miss Rate}_m \quad (24)$$

where $\mathbb{I}(\cdot)$ is the indicator function.

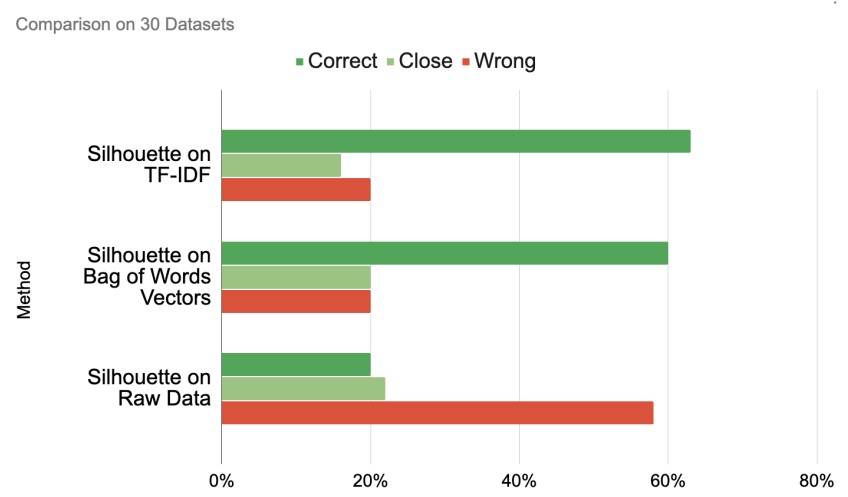

Figure 1: Comparing Results

### 5.1 *Baseline Results: Raw Time Series Data*

For each dataset $D_i$, we computed the Silhouette score $S(K, D_i)$ for $K = 2, ..., K_{\max}$, predicting:

$$K_{\text{pred,raw}}^i = \arg \max_K S(K, D_i) \tag{25}$$

Experiments on 30 datasets from the UCR archive yielded:

$$\text{Accuracy}_{\text{raw}} \approx 0.20 \tag{26}$$
$$\text{Near-miss Rate}_{\text{raw}} \approx 0.22 \tag{27}$$
$$\text{Error Rate}_{\text{raw}} \approx 0.58 \tag{28}$$

These results indicate that applying the Silhouette Coefficient directly to raw time series data yields suboptimal performance, underscoring the need for more sophisticated representations for clustering purposes.

### 5.2 *BoW Vector Results*

For the Bag-of-Words approach, we generated vectors from SAX words of the time series. Let $w$ and $\alpha$ be the SAX window size and alphabet size, respectively. For each dataset $D_i$, we optimized:

$$(w_i, \alpha_i, K_{\text{pred,BoW}}^i) = \arg \max_{w, \alpha, K} S_{\text{BoW}}(K, D_i, w, \alpha) \tag{29}$$

where $S_{\text{BoW}}$ is the Silhouette score computed on BoW vectors. This optimization was performed over ranges $w \in [w_{\min}, w_{\max}]$ and $\alpha \in [\alpha_{\min}, \alpha_{\max}]$. The performance metrics for the BoW approach were:

$$\text{Accuracy}_{\text{BoW}} \approx 0.60 \tag{30}$$
$$\text{Error Rate}_{\text{BoW}} \approx 0.20 \tag{31}$$

### 5.3 *TF-IDF Vector Results*

For TF-IDF vectors, we introduced additional parameters $\theta_{\min}$ and $\theta_{\max}$ for minimum and maximum word frequencies. The optimization problem became:

$$(K_{\text{pred,TF-IDF}}^i, \theta_{\min,i}, \theta_{\max,i}) = \arg \max_{K, \theta_{\min}, \theta_{\max}} S_{\text{TF-IDF}}(K, D_i, w_i, \alpha_i, \theta_{\min}, \theta_{\max}) \tag{32}$$

where $w_i$ and $\alpha_i$ are the optimal SAX parameters from the BoW approach.

### 5.4 *Comparison*

Let $\mathcal{A} = \text{raw}, \text{BoW}, \text{TF-IDF}$ be the set of approaches. We compare their performance using:

$$\text{Relative Improvement}_m = \frac{\text{Accuracy}_m - \text{Accuracy}_{\text{raw}}}{\text{Accuracy}_{\text{raw}}} \times 100\% \tag{33}$$

for $m \in \text{BoW}, \text{TF-IDF}$. The results show:

$$\text{Relative Improvement}_{\text{BoW}} \approx 200\% \tag{34}$$
$$\text{Relative Improvement}_{\text{TF-IDF}} \approx 205\% \tag{35}$$

These results indicate a strong correlation between the optimal number of clusters and the BoW/TF-IDF vector representations of time series data, significantly outperforming the raw data approach.

## 6 CONCLUSION AND FUTURE WORKS

This research presents a novel approach to time series clustering by extending the Symbolic Pattern Forest algorithm Li et al. (2019a) to predict the optimal number of clusters $K^*$ without prior knowledge. This addresses a fundamental limitation in existing clustering methods, which typically require the number of clusters to be specified in advance. Our approach can be formalized as:

$$K^* = \arg\max_K S(K, \mathbf{v}(\mathbf{X}i)i = 1^N) \tag{36}$$

where $S$ is the Silhouette Score, $\mathbf{v}$ is our vector representation (BoW or TF-IDF), and $\mathbf{X}ii = 1^N$ is the dataset. Unlike traditional methods such as K-means, K-shape, or spectral clustering, which solve:

$$\mathcal{C}^* = \arg\min_{\mathcal{C}} \sum_{i=1}^K \sum_{\mathbf{X} \in C_i} d(\mathbf{X}, \boldsymbol{\mu}_i) \tag{37}$$

for a fixed $K$, our method determines $K^*$ automatically. We demonstrate that while the Silhouette Score $S(K)$ is ineffective on raw time series data $\mathbf{X} = (x_1, ..., x_T)$, it becomes highly effective when applied to our SAX-based vector representations:

$$\mathbf{v}\text{BoW}(\mathbf{X}) = f\text{BoW}(\text{SAX}(\mathbf{X}, w, \alpha)) \quad \mathbf{v}\text{TF-IDF}(\mathbf{X}) = f\text{TF-IDF}(\text{SAX}(\mathbf{X}, w, \alpha), \theta_{\min}, \theta_{\max}) \tag{38}$$

This innovative approach significantly outperforms baseline methods, offering the first reliable technique for time series clustering without prior knowledge of cluster numbers. Future work will extend our experiments to all 128 datasets in the UCR archive Dau et al. (2019a). We aim to further refine our methodology to enhance prediction accuracy:

$$\text{Accuracy} = \frac{1}{|\mathcal{D}|} \sum_{D \in \mathcal{D}} \mathbb{I}(K^* D = K\text{true}, D) \tag{39}$$

By eliminating the need for a priori cluster number specification, our method opens new possibilities in exploratory data analysis and unsupervised learning for time series. It provides a more robust and flexible approach to uncovering inherent structures in time series data, potentially leading to discoveries that might be missed by traditional fixed-K methods. Our ongoing work thus contributes significantly to time series analysis by providing the first truly unsupervised clustering methodology for time series data. This advancement is crucial for applications where the underlying number of clusters is unknown or may vary, such as in anomaly detection, pattern discovery, and dynamic system analysis.

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

## A  DATASET-SPECIFIC RESULTS

Table 1: Silhouette on Raw Time Series

| Dataset | Actual Cluster | Predicted | Remarks |
|---|---|---|---|
| Beef | 5 | 3 | Wrong |
| FaceFour | 4 | 3 | Close |
| Fish | 7 | 2 | Wrong |
| GunPoint | 2 | 2 | Correct |
| Rock | 4 | 2 | Wrong |
| HouseTwenty | 2 | 4 | Wrong |
| EthanolLevel | 4 | 2 | Wrong |
| Wine | 2 | 2 | Correct |
| Wafer | 2 | 2 | Correct |
| SyntheticControl | 6 | 3 | Wrong |
| InlineSkate | 7 | 2 | Wrong |
| InsectEPGRegularTrain | 3 | 2 | Close |
| GunPointAgeSpan | 2 | 8 | Wrong |
| Haptics | 5 | 2 | Wrong |
| UMD | 3 | 2 | Close |
| Symbols | 6 | 3 | Wrong |
| OliveOil | 4 | 2 | Wrong |
| HandOutlines | 2 | 2 | Correct |
| Meat | 3 | 2 | Close |
| ECG200 | 2 | 3 | Close |
| WormsTwoClass | 2 | 4 | Wrong |
| Worms | 5 | 4 | Close |
| Plane | 7 | 6 | Close |
| Strawberry | 2 | 2 | Correct |
| Trace | 4 | 2 | Wrong |
| Lightning7 | 7 | 5 | Wrong |
| MoteStrain | 2 | 4 | Wrong |
| ChinaTown | 2 | 2 | Correct |
| TwoPatterns | 4 | 2 | Wrong |
| TwoLeadECG | 2 | 4 | Wrong |

Table 2: Silhouette on BoW Vectors

| Dataset | SAX Window | SAX Alphabet | Actual | Predicted | Remarks |
|---|---|---|---|---|---|
| Beef | 5 | 8 | 5 | 5 | Correct |
| FaceFour | 40 | 8 | 4 | 3 | Close |
| Fish | 5 | 20 | 7 | 8 | Close |
| GunPoint | 5 | 9 | 2 | 2 | Correct |
| Rock | 8 | 5 | 4 | 4 | Correct |
| HouseTwenty | 50 | 8 | 2 | 2 | Correct |
| EthanolLevel | 350 | 10 | 4 | 3 | Close |
| Wine | 3 | 4 | 2 | 2 | Correct |
| Wafer | 20 | 5 | 2 | 2 | Correct |
| SyntheticControl | 3 | 6 | 6 | 6 | Correct |
| InlineSkate | 100 | 8 | 7 | 2 | Wrong |
| InsectEPGRegularTrain | 100 | 3 | 3 | 2 | Close |
| GunPointAgeSpan | 10 | 8 | 2 | 2 | Correct |
| Haptics | 20 | 10 | 5 | 2 | Wrong |
| UMD | 20 | 8 | 3 | 3 | Correct |
| Symbols | 30 | 10 | 6 | 3 | Wrong |
| OliveOil | 100 | 4 | 4 | 3 | Close |
| HandOutlines | 100 | 4 | 2 | 2 | Correct |
| Meat | 20 | 4 | 3 | 2 | Close |
| ECG200 | 100 | 4 | 2 | 2 | Correct |
| WormsTwoClass | 200 | 10 | 2 | 2 | Correct |
| Worms | 200 | 10 | 5 | 2 | Wrong |
| Plane | 10 | 4 | 7 | 7 | Correct |
| Strawberry | 50 | 4 | 2 | 2 | Correct |
| Trace | 50 | 4 | 4 | 4 | Correct |
| Lightning7 | 100 | 8 | 7 | 4 | Wrong |
| MoteStrain | 5 | 4 | 2 | 2 | Correct |
| ChinaTown | 12 | 3 | 2 | 2 | Correct |
| TwoPatterns | 50 | 4 | 4 | 2 | Wrong |
| TwoLeadECG | 40 | 8 | 2 | 2 | Correct |

Table 3: Silhouette on TF-IDF Vectors

| Dataset | SAX Window | SAX Alphabet | Min Freq | Max Freq | Actual | Predicted | Remarks |
|---|---|---|---|---|---|---|---|
| Beef | 5 | 8 | 0.01 | 0.9 | 5 | 5 | Correct |
| FaceFour | 40 | 8 | 0.1 | 0.9 | 4 | 3 | Close |
| Fish | 5 | 20 | 0.001 | 0.01 | 7 | 8 | Close |
| GunPoint | 5 | 9 | 0.01 | 0.9 | 2 | 2 | Correct |
| Rock | 8 | 5 | 0.01 | 0.9 | 4 | 4 | Correct |
| HouseTwenty | 50 | 8 | 0.01 | 0.9 | 2 | 2 | Correct |
| EthanolLevel | 350 | 10 | 0.001 | 0.99 | 4 | 3 | Close |
| Wine | 3 | 4 | 0.1 | 0.9 | 2 | 2 | Correct |
| Wafer | 20 | 5 | 0.001 | 0.99 | 2 | 2 | Correct |
| SyntheticControl | 3 | 6 | 0.1 | 0.9 | 6 | 6 | Correct |
| InlineSkate | 100 | 8 | 0.15 | 0.8 | 7 | 2 | Wrong |
| InsectEPGRegularTrain | 100 | 3 | 0.25 | 0.9 | 3 | 2 | Close |
| GunPointAgeSpan | 10 | 8 | 0.001 | 0.9 | 2 | 2 | Correct |
| Haptics | 20 | 10 | 0.2 | 0.95 | 5 | 2 | Wrong |
| UMD | 20 | 8 | 0.001 | 0.99 | 3 | 3 | Correct |
| Symbols | 30 | 10 | 0.001 | 0.99 | 6 | 3 | Wrong |
| OliveOil | 100 | 4 | 0.1 | 0.9 | 4 | 3 | Close |
| HandOutlines | 100 | 4 | 0.1 | 0.9 | 2 | 2 | Correct |
| Meat | 20 | 4 | 0.1 | 0.9 | 3 | 2 | Close |
| ECG200 | 100 | 4 | 0.1 | 0.9 | 2 | 2 | Correct |
| WormsTwoClass | 200 | 10 | 0.1 | 0.9 | 2 | 2 | Correct |
| Worms | 200 | 10 | 0.1 | 0.9 | 5 | 2 | Wrong |
| Plane | 10 | 4 | 0.01 | 0.99 | 7 | 7 | Correct |
| Strawberry | 50 | 4 | 0.01 | 0.99 | 2 | 2 | Correct |
| Trace | 50 | 4 | 0.01 | 0.99 | 4 | 4 | Correct |
| Lightning7 | 100 | 80 | 0.01 | 0.99 | 7 | 4 | Wrong |
| MoteStrain | 5 | 4 | 0.01 | 0.99 | 2 | 2 | Correct |
| ChinaTown | 12 | 3 | 0.01 | 0.99 | 2 | 2 | Correct |
| TwoPatterns | 50 | 4 | 0.01 | 0.99 | 4 | 2 | Wrong |
| TwoLeadECG | 40 | 8 | 0.01 | 0.99 | 2 | 2 | Correct |