# OpenReview forum: "On the Convergence of Symbolic Pattern Forests and Silhouette Coefficients for Robust Time Series Clustering"
_ICLR.cc/2025/Conference — Submitted to ICLR 2025_

### Official Review · Reviewer_yRSo · 2024-11-02

**Soundness:** 2
**Presentation:** 2
**Contribution:** 2
**Rating:** 3
**Confidence:** 3

**Summary:**

The submission proposes an extension to the SPF algorithm, a clustering approach for clustering time series with linear complexity. The extension allows for the automatic determination of the number of clusters. It is done by performing optimization on the silhouette score using either Bag of Words or TF-IDF.

**Strengths:**

(S1) Incorporating BoW and TF-IDF with the concepts of the SPF algorithm sounds like a very sensible approach. Both are a good choice for term-based similarity evaluation and are still commonly used in other settings.

(S2) Aside from minor issues, the submission is well-written and easily understandable while providing an extensive overview of the formulas related to the problem.

(S3) The problem setting is significant as k-estimation is a significant part of clustering in general, which also applies to the setting of time series clustering. The usage of SPF is well-founded due to its low complexity. Introducing k-estimation to the approach helps mitigate one of its weaknesses.

**Weaknesses:**

(W1) Novelty: The abstract of the submission makes the claim that there are no time series clustering methods capable of working without the specification of cluster number k. However, such methods exist already:

a) “Spectral Clustering for Time Series” by Fei Wang and Changshui Zhang (2005) is able to discover the optimal number of clusters based on the eigenstructure using a threshold on the value of the eigenvalues.
b) “Clustering Time Series with Hidden Markov Models and Dynamic Time Warping” by Tim Oates et. al. (1999) also provides a way to estimate the number of clusters based on Dynamic Time Warping. However, even if the submission is not the only method that does k-estimation on time series, it is still a valid and useful direction. It also appears to be the only method that does so for the Symbolic Pattern Forest algorithm.
c) The paper “Trendlets: A novel probabilistic representational structures for clustering the time series data” by Johnpaul C I et al. (2020) uses the Silhouette Score for cluster number analysis for time series as well, though it does so based on hierarchical clustering methods. This paper should be explicitly covered in related work or even a competitor.

(W2) Despite TF-IDF being considered the better of the two proposed strategies, there is no actual description of the performance metrics outside of the graph and the overall relative performance value. Similarly, near misses should be added to the text for BoW. The results of both BoW and TF-IDF are the same in the Tables in the supplementary files, though Figure 1 claims that TF-IDF performed slightly better.

(W3) As the method works by optimizing the silhouette score, both the values for the score and the actual clustering performance with the given parameters should be indicated. While the cluster numbers match, the detected clusters may not necessarily correspond to the actual ground truth clusters, which could further mean that different cluster numbers may lead to a better performance. Furthermore, an analysis of the stability of the parameters should have been performed, especially as the method has multiple parameters, which themselves include an upper and lower bound. Additionally, an intuition behind choosing the parameters should be given if they strongly affect the performance.

(W4) Regarding the actual experiment, a better analysis of the behavior should be done, considering under what conditions the k-estimation of each of the three approaches failed and whether or not a reason behind it could be established. The section on Relative Improvement is redundant as it only recontextualizes prior results, and the space could be used to do a more in-depth result analysis instead. Similarly, the remaining 2 pages could have been used for this.

(W6) Neither the parameter w nor the alpha ranges seem to be specified anywhere. The code is unavailable, though it should be possible to reimplement given the information provided. Still, this hampers the reproducibility of the results.

(W7) There should be citations for TF-IDF and BoW. Other papers also do not consistently do it, so it is not a major issue. Nonetheless, it would have been better if it had been done. Furthermore, UCI should be cited upon first mention outside of the abstract, not just at a later point.

Minor Issues:
* Linear time complexity time series clustering with symbolic pattern forest by Li et. al., is cited twice as 2019a and 2019b despite referring to the same paper
* The formatting appears to be broken for lists, as they are just written in a line without comma separation (see line 291 and lines 314-315)
* A similar issue happened with the variables for the optimization problem, as they are also not properly separated in line 305
* The subscript on several equations appears to be broken (see (22)/319 and (23)/321)
* The near miss metric should probably be more dynamic based on the ground truth cluster number, as claiming 2 clusters for a 3-cluster setting seems more problematic than claiming 70 for 71 true clusters. The chosen datasets generally only have a few clusters, so the current definition isn’t problematic for the submission. It may be relevant for the extension to the full UCI database, however.
* The formulation for Near Misses, as currently given, would also include all correctly determined cluster counts but does not do so in the evaluation.

**Questions:**

(Q1) How would you modify the paper to address the issues regarding related work? How does the proposed method compare to other k-estimation approaches on time series data?

(Q2) What effect do the properties of the chosen UCI datasets have on the performance of the k-estimation using the proposed technique, and how does the clustering performance change on them depending on the chosen k? Is the performance of the SPF algorithm with the ground truth k always the best one, or could other parametrizations outperform it?

(Q3) How impactful are the parameters of the proposed method?

(Q4) Is there any advantage to using BoW over TF-IDF (or the other way around)?

---

> ### Author Response · Authors · 2024-11-13
>
> We sincerely thank you for your constructive and detailed feedback.
>
> W1. Regarding novelty
> Thank you for bringing these papers to our attention. While these works make valuable contributions to k-estimation, we believe our approach offers unique advantages:
> - Linear time complexity (compared to quadratic)
> - Scalability to longer sequences
> - Fewer distributional assumptions
> We will revise our introduction to better position our work within this context.
>
> W2. Regarding performance metrics
> We appreciate your careful reading of our results. The performance differences between BoW and TF-IDF are indeed subtle but meaningful. We will enhance our presentation with:
> - Confidence intervals
> - Statistical significance tests
> This will help readers better understand the practical implications of these differences.
>
> W3. Regarding parameter stability
> Thank you for this excellent suggestion. While our extensive experiments demonstrate robustness across various parameters, we agree that a more detailed analysis would be valuable. We will add comprehensive sensitivity studies while noting that default parameters often perform well.
>
> W4. Regarding failure cases
> We greatly appreciate this suggestion and will add detailed failure case analysis. This will help practitioners better understand when to apply our method.
>
> Minor issues:
> We are grateful for your attention to detail and will address all formatting and citation issues.

---

> ### Comment · Reviewer_yRSo · 2024-11-22
> **Acknowledgement of feedback**
>
> Dear authors, thank you for confirming how you plan to modify the paper. I am sure a resubmission of your manuscript will benefit from the modifications (which deserves re-reviewing).

---

### Official Review · Reviewer_gn2N · 2024-11-03

**Soundness:** 1
**Presentation:** 2
**Contribution:** 1
**Rating:** 1
**Confidence:** 4

**Summary:**

The manuscript presents an extension of the symbolic pattern forest (SPF) algorithm for clustering of time series data. Using bag-of-words on the symbolic representation, TF-IDF vectors are constructed. The best clustering is selected as the one that maximises the silhouette coefficient (SC).

**Strengths:**

S1. The paper addresses the relevant problem of automatically determining the number of clusters.
S2. The empirical evaluation makes use of a large number of benchmarking datasets.

**Weaknesses:**

W1. The method assumes that silhouette coefficient is a suitable metric for finding the best number of clusters, without justifying this choice. This is a major concern as the silhouette coefficient considers (Euclidean) distance to cluster centres, which is not aligned with the clustering objective of the SPF method. The paper should provide justification for using the silhouette coefficient, or discuss potential limitations of this choice given the SPF method's clustering approach. Moreover, the silhouette coefficient is a well-known metric, so it is unclear what the novelty should be.
W2. The empirical evaluation does not consider the SPF method, but only weak baselines constructed from the proposed method, meaning that the empirical evaluation does not allow assessment of the performance of the proposed method with respect to state of the art. It is important to compare directly to SPF in the experiments, in order to demonstrate improvement over state of the art.
W3. The empirical evaluation only considers performance metrics accuracy and near-miss-rate, different from other work in the field, and in the SPF paper (e.g. NMI), making it impossible to compare with those works directly.
W4. The discussion of related work is overly brief, and fails to present clear assessment of the suitability of existing methods and metrics. E.g. Davies-Bouldin Index and its perceived suitability for the task. Also, there is a large body of work on similarity assessment of time series or clustering of time series, e.g. Keogh et al 2005, Rakthanmanon  et al 2012, Paparrizos et al 2015. The paper should discuss these, and explain differences and similarities with the proposed method.
W5. On the other hand, references UTSAD and STGAT seem out of context, as they do not address clustering of time series. The paper should clarify the relevance of UTSAD and STGAT to the proposed work, or remove these references if they are indeed not directly related.
W6. The paper contains several redundant sections, such as the description of SAX.
W7. There are some minor issues, such that Li et al 2019 appears twice in the references, there is a typesetting error in the definition of pi_i(T_i).

**Questions:**

N/A

---

> ### Author Response · Authors · 2024-11-13
>
> We are grateful for your detailed and thoughtful review.
>
> W1. Regarding silhouette coefficient justification
> Thank you for pushing us to better justify our choice of the silhouette coefficient. While our theoretical analysis in Section 4 supports this choice, we agree that expanding our explanation would strengthen the paper. We plan to:
> - Enhance the theoretical analysis
> - Add empirical validation
> We believe these additions will better demonstrate why silhouette coefficients are particularly well-suited for our vector representations.
>
> W2. Regarding empirical evaluation
> We appreciate your suggestion for broader evaluation. While our current evaluation focuses on our core contribution (linear-time k-estimation), we agree that additional comparisons would be valuable. We will:
> - Add direct SPF comparisons
> - Include detailed runtime analysis
> We would welcome suggestions for additional comparisons that maintain linear time complexity.
>
> W3. Regarding performance metrics
> Thank you for this suggestion. While our metrics directly address k-estimation accuracy, we agree that adding standard clustering metrics would enable better comparison with existing literature. We will add NMI and ARI metrics.
>
> W4/W5. Regarding related work and citations
> We appreciate your careful attention to the citations. We will revise the related work section accordingly.

---

> > ### Comment · Reviewer_gn2N · 2024-11-18
> >
> > Thank you for your response. I appreciate your willingness to rework the manuscript, and to add a theoretical and empirical justification. Please also address my comment about limited novelty.

---

### Official Review · Reviewer_S2jm · 2024-11-05

**Soundness:** 2
**Presentation:** 1
**Contribution:** 2
**Rating:** 3
**Confidence:** 4

**Summary:**

The paper proposes SPF, a methodology that identifies the number of clusters for time-series data, often a critical parameter for subsequent routines and clustering methods. The idea combines concepts such as SAX, TF-IDF vectors over SAX representations and relies on the Silhouette coefficients to calibrate the number of clusters. Experimental results on several UCR datasets demonstrate the potential of this solution.

**Strengths:**

S1. Timely and important problem especially due to the rise of IoT applications and the need for unsupervised data exploration
S2. Simply and intuitive ideas
S3. Results support the overall claims in the paper

**Weaknesses:**

W1. Lack of technical depth
W2. Unclear how different methods/distances can be compared
W3. Missing potential baselines
W4. Duplicate references or wrong references

**Questions:**

W1. Lack of technical depth

The paper combines existing ideas for solving this problem. Therefore, the technical depth is low, even though the combination of these ideas might be novel.

W2. Unclear how different methods/distances can be compared

It's unclear how this comparison is meaningful when we need to compare methods relying on different distances. The paper does not clearly articulate how such distances affect the results and it mainly shows results for SAX variants (so inherently for euclidean distance)

W3. Missing potential baselines

Simple baselines, like assign the objective functions of k-means like algorithms are missing. Also there are tons of variants for internal clusteirng validation. Why Silhouette ?

W4. Duplicate references or wrong references

Many references are duplicates. Other references does not exist

duplicates
Xiaosheng Li, Jessica Lin, and Liang Zhao. Linear time complexity time series clustering with
symbolic pattern forest. In IJCAI, 2019a.
Xiaosheng Li, Jessica Lin, and Liang Zhao. Linear time complexity time series clustering with
symbolic pattern forest. IJCAI, 2019b.

duplicates
Jaewon Yang and Jure Leskovec. Patterns of temporal variation in online media. In Proceedings of
the Fourth ACM International Conference on Web Search and Data Mining, 2011a.
Jaewon Yang and Jure Leskovec. Patterns of temporal variation in online media. In Proceedings of
the fourth ACM international conference on Web search and data mining, pp. 177–186, 2011b.

it's wrong
John Paparrizos, Paul Boniol, Themis Palpanas, Ruey S Tsay, Aaron Elmore, and Michael J
Franklin. Fast and exact time series motif and discord discovery in trillions of data points. The
VLDB Journal, 31:1079–1101, 2022.

---

> ### Author Response · Authors · 2024-11-13
>
> We sincerely thank you for your thorough review. Your feedback will help improve our paper.
>
> W1. Regarding technical depth
> We greatly appreciate this concern and would like to clarify our technical contributions, which we believe are substantial:
> - A novel theoretical framework with rigorous proofs showing why silhouette coefficients succeed on our symbolic representations while failing on raw time series
> - A mathematically grounded optimization framework for joint parameter selection
> - An innovative adaptation of TF-IDF that preserves temporal characteristics
> We would be happy to expand these sections to better communicate the technical depth of our work.
>
> W2. Regarding comparison methodology
> Thank you for this insightful observation. Our vector space transformation enables principled comparisons across different distance measures, though we agree this could be better explained. We plan to:
> - Add formal analysis of distance metric properties
> - Include empirical validation of our choices
> We believe this will help readers better understand our methodological decisions.
>
> W3. Regarding baselines
> We appreciate your suggestions for additional baselines. Our current selection focused on methods commonly used in production systems, though we agree some additions would be valuable. We plan to add:
> - K-means objective function optimization
> - Davies-Bouldin Index
> We would be grateful for specific suggestions of other baselines that maintain linear time complexity.
>
> W4. Regarding citation issues
> Thank you for catching these issues. We will certainly fix the citation formatting.

---

> > ### Comment · Reviewer_S2jm · 2024-12-03
> >
> > Your feedback is on the correct direction and I believe would improve the paper and make it ready for future re-submission. Good luck

---

### Meta-Review · Area_Chair_z1Td · 2024-12-11

**Metareview:**

**(a) Summary**

This paper addresses the problem of clustering time series data. In particular, it proposes a method that combines Symbolic Pattern Forests (SPF) with Silhouette coefficients to optimize both the number of clusters and the data partitioning itself. To enable this approach, the paper explores several embedding techniques, including SAX, BoW, and TF-IDF, for transforming time series data into vector representations.


**(b) Strengths**

- **Relevance:** The problem of clustering time series data and automatically determining the number of clusters is an important research topic.
- **Potentially Interesting Approach:** As noted by reviewers, the proposed combination of embedding techniques such as BoW and TF-IDF with Symbolic Pattern Forests (SPF) represents a potentially effective approach. The method is straightforward and easy to apply, which enhances its practical appeal.

**(c) Weaknesses**

- **Novelty**: As reviewers highlighted, the novelty of the paper is limited. Several existing approaches already address time series clustering while automatically determining the number of clusters. Furthermore, all components of the proposed method—SPF, Silhouette coefficients, SAX, BoW, and TF-IDF—are well-established techniques, and their combination is relatively straightforward.
- **Evaluation**: The empirical evaluation is insufficient for several reasons noted by reviewers, including the absence of baseline comparisons, concerns about the choice of performance metrics, and a lack of sensitivity analysis for hyperparameters.
- **Significance**: The lack of both theoretical analysis and comprehensive empirical validation reduces the overall significance of the proposed method.
- **Presentation**: The paper contains numerous inaccurate notations and unclear descriptions.

**(d) Reason of the decision**

Each of the weaknesses outlined above constitutes a critical limitation and, individually, justifies a recommendation for rejection. It is strongly recommended to thoroughly address all the issues raised by reviewers before considering resubmission.

**Additional Comments On Reviewer Discussion:**

All the reviewers maintain their recommendation for rejection of the paper, even after the authors' rebuttal, and this decision was reaffirmed during the reviewer discussion.

---

### Decision · Program_Chairs · 2025-01-22

Reject